# Mapping the existing body of knowledge on new and repurposed TB vaccine implementation: A scoping review

**Joeri S. Buis**[1], **Degu Jerene**[1]*, **Agnes Gebhard**[1], **Roel Bakker**[1], **Arman Majidulla**[2], **Andrew D. Kerkhoff**[3], **Rupali J. Limaye**[2], **Puck T. Pelzer**[1]

**1** KNCV Tuberculosis Foundation, The Hague, The Netherlands, **2** Department of International Health, Bloomberg School of Public Health, Johns Hopkins University, Baltimore, Maryland, United States of America, **3** Division of HIV, Infectious Diseases and Global Medicine Zuckerberg San Francisco General Hospital and Trauma Center, Center for Tuberculosis, University of California San Francisco, San Francisco, California, United States of America

* Degu.dare@kncvtbc.org

**Data Availability Statement:** This is a review of published (preprint) research articles. All studies reviewed are available in either of the following

## Abstract

There is global consensus on the urgent need for a safe and effective TB vaccine for adults and adolescents to improve global TB control, and encouragingly, several promising candidates have advanced to late-stage trials. Significant gaps remain in understanding the critical factors that will facilitate the successful implementation of new and repurposed TB vaccines in low- and middle-income countries (LMICs), once available. By synthesizing the existing body of knowledge, this review offers comprehensive insights into the current state of research on implementation of these adult and adolescent vaccines. This review explores four key dimensions: (1) epidemiological impact, (2) costing, cost-effectiveness, and/or economic impact, (3) acceptability, and the (4) feasibility of implementation; this includes implementation strategies of target populations, and health system capabilities. Results indicate that current research primarily consists of epidemiological and costing/cost-effectiveness/economic studies in India, China, and South Africa, mainly modelling with M72/AS01, BCG revaccination, and hypothetical vaccines. Varying endpoints, vaccine efficacies, and vaccination coverages were used. Globally, new, and repurposed TB vaccines are estimated to save millions of lives. Economically, these vaccines also demonstrate promise with expected cost-effectiveness in most countries. Projected outcomes were dependent on vaccine characteristics, target population, implementation strategy, timing of roll out, TB burden/country context, and vaccination coverage. Potential barriers for vaccine acceptability included TB-related stigma, need for a second dose, and cost, while low pricing, community and civil society engagement and heightened public TB awareness were potential enablers in China, India, and South Africa. Potential implementation strategies considered spanned from mass campaigns to integration within existing vaccine programs and the primary target group studied was the general population, and adults and adolescents. In conclusion, future research must have broader geographical representations to better understand what is needed to inform tailored vaccine programs to accommodate diverse country contexts and population groups to achieve optimal implementation and impact. Furthermore, this review

databases: PubMed, Medrxiv, and PLOS. The lists of search strings can be found in S1 Text. Additionally, all relevant data used in the review are in the manuscript and supplementary tables.

**Funding:** This review is executed under the SMART4TB project. SMART4TB is made possible by the generous support of the American people through the United States Agency for International Development (USAID) and is implemented under cooperative agreement number 7200AA20CA00005. The consortium is managed by prime recipient, Johns Hopkins University.

**Competing interests:** The authors have declared that no competing interests exist.

underscores the scarcity of research on acceptability of new and repurposed TB vaccines and their delivery among potential beneficiaries, the most promising implementation strategies, and the health system capabilities necessary for implementation. The absence of this knowledge in these areas emphasizes the crucial need for future research to ensure effective TB vaccine implementation in high burden settings worldwide.

## Introduction

Tuberculosis (TB) remains a persistent global health challenge, imposing a significant burden on communities worldwide, despite substantial health efforts, including the availability of effective treatment, and widespread use of the Bacillus Calmette–Guerin (BCG) vaccine. In 2022 alone, 10.6 million people fell ill with TB, with approximately 1.3 million deaths [1]. Of all reported TB patients, 99% were in low- and middle-income countries (LMICs) [2]. India, Indonesia, China, the Philippines, Pakistan, Nigeria, Bangladesh, and the Democratic Republic of Congo, account for approximately two-thirds of all patients worldwide. The development of new and repurposed tuberculosis (TB) vaccines is a promising approach with the potential to transform TB prevention, aiding (LMICs) in achieving the World Health Organization (WHO)'s End TB targets [1]. The WHO has identified new and repurposed vaccines as a cornerstone of the WHO End TB Strategy, with the ultimate goal to build a world without TB [1]. Effective implementation will be pivotal for maximizing public health impact once new and/or repurposed vaccines are available.

Currently, the BCG vaccine is the only licensed vaccine against TB. First employed in 1921, the vaccine is one of the most widely used vaccines globally, with 80% of all countries recommending BCG vaccination for neonates [3], and a coverage of >80% on average. BCG vaccination primarily prevents severe forms of TB in young children and multiple limitations have been identified: varying effectiveness, contraindication in people who are living with HIV/AIDS (PLHIV), and the lack of protection against reactivation of TB. Additionally, there is little evidence that it prevents the development of TB in adults and adolescents [4]. These are important limitations as adults and adolescents account for about 90% of TB incidence and TB is the leading cause of death from a single infectious agent in PLHIV [1, 5].

To fill the critical research and development gap in TB vaccines, there has been an increase in clinical trials evaluating the effectiveness and safety of TB vaccine candidates that target adults and adolescents. Approximately 17 TB vaccine candidates are in the clinical development pipeline, of which six have progressed to phase 3 development and are intended for use in adults and adolescents [5, 6]. These six vaccines—MIP, VMP1002, GamTBvac, MTBVAC, BCG (re)vaccination, and M72/AS01E—are being tested for their potential to prevent disease (PoD) or prevent infection (PoI) [7]. Of these, promising phase 2b results were published for the M72/AS01E vaccine, revealing a vaccine efficacy of 49.7% (90% confidence interval [CI] = 12.1 to 71.2 and 95% CI = 2.1 to 74.2) against development of TB disease over a three year follow-up period [6]. The BCG revaccination is being repurposed for adults and adolescents and efficacy was assessed, in adolescents between the ages of 12–17 years. Results indicated an estimated vaccine efficacy (VE) of 45.4% (95% CI, 6.4 to 68.1) for PoI [8]. While clinical trials are instrumental in evaluating the safety and efficacy of TB vaccine candidates, evidence on vaccine implementation beyond the clinical trial setting is critical to achieving public health impact. There is an urgent need to understand how to effectively implement and scale-up

vaccination programs to achieve optimal public health impact, following the approval of new and repurposed TB vaccines.

In the last few years, evidence has accumulated regarding the potential impact of new and repurposed TB vaccines. Modelling studies have shown that a new TB vaccine with an efficacy of at least 50% for preventing the development of TB disease, a faster and broader impact could be achieved targeting adults and adolescents rather than infants and children [9]. However, Harris et al. argue that the assumptions used in the modelling—such as having separate vaccines specifically licensed for people who are not yet infected (pre-infection vaccine) or have been infected (post-infection vaccine)—may not be feasible. This infeasibility arises from the need to screen for TB infection prior to vaccination, which incurs high costs and relies on tuberculin skin tests or IGRA tests. These tests are not only expensive but also have limitations in accurately diagnosing TB infection status, thus complicating targeted vaccination efforts [9].

Two recent documents, the Global Roadmap for Research and Development for TB Vaccines and WHO's Evidence Considerations for Vaccine Policy (ECVP), identify key actions and considerations to prepare for the effective implementation of a new and repurposed TB vaccine, once approved [10, 11]. Crucial TB vaccine implementation components to realizing successful TB vaccination programs for adults and adolescents in LMICs include understanding the TB burden, the public health and economic impact of TB vaccination in target populations, developing and implementing strategic plans for TB vaccination, strengthening healthcare capacity and infrastructure, enhancing public awareness, and increasing acceptance among the public [10, 11].

A comprehensive review and synthesis of the existing literature on best practices regarding acceptability and feasibility of implementing new or repurposed TB vaccines for adults and adolescents has not yet been undertaken, nor has an overview of projected epidemiological impact and costing, cost-effectiveness and economic impact. This work would offer critical insights to understanding what will be needed to prepare for and contribute to the effective launch and scalable delivery of a licensed TB vaccine upon availability, including the identification of knowledge gaps.

Therefore, the objective of this scoping review is to assess and synthesize existing research and evidence on TB vaccine implementation in adults and adolescents in LMICs, regarding projected epidemiological impact, economic impact, acceptability, and feasibility, and identify priority areas for future research to prepare for deployment of new and repurposed TB vaccines. The specific questions included:

What is the existing body of knowledge on the:

1. epidemiological impact of new and repurposed TB vaccines targeting adults and/or adolescents as estimated by modelling studies?

2. costing, cost-effectiveness, and/or economic impact of new and repurposed TB vaccines targeting adults and/or adolescents as estimated by modelling studies?

3. acceptability of new and repurposed TB vaccines targeting adults and/or adolescents?

4. implementation feasibility of vaccination programs using new and repurposed TB vaccines, targeting adults and/or adolescents?

    a. the potential implementation strategies for new and repurposed TB vaccines?

    b. the existing body of knowledge on health system readiness for the implementation and scale up of a new and repurposed TB vaccine?

## Methods

### Study design overview

We employed a narrative synthesis approach [12], sourcing literature from PubMed, Medrxiv, PLOS journals, and expert suggestions. Articles were selected using a systematic method, and data extraction and analysis were carried out iteratively using the Roadmap [10] and ECVP [11].

### Search methods

The narrative synthesis approach involved three systematic search components: (1) database search, (2) search on internet sites, and (3) identification of relevant articles by the team (JSB or PTP). The search strings for the database queries combined the following concepts: TB, TB vaccination, and vaccine preparedness. A combination of Medical Subject Headings (MeSH) terms and separate keywords relating to the research questions were included to create a broad and inclusive search strategy. All concepts included variations of the keywords. Filtering removed non-human, infant/children, clinical, and immunological studies, limiting to English articles published between 2013–2023. Using the search strings, article retrieval was carried out on 30th April 2023. The detailed search strings are provided in S1 Text.

1. Databases: Searches were conducted using PubMed and Medrxiv.

2. Internet sites: PLOS journals were identified as having relevant literature related to this topic, however not all PLOS journals were indexed in PubMed or Medrxiv at the time of conducting the search. Therefore, a quick literature search was conducted within the PLOS journal.

3. Identification of additional literature: to accommodate the identification of articles currently not covered within the above search components, additional literature identified by the authors were also included.

### Inclusion and exclusion criteria

Scientific literature (including preprints) related to the implementation of a new and repurposed TB vaccines for adults and adolescent was included (Table 1). We focused on studies conducted in or concentrating on LMICs and we focused on articles published between 2013 and 2023. Expanding the timeframe was not anticipated to yield additional relevant articles, given that the development of TB vaccine candidates has been a more recent occurrence.

**Table 1. Inclusion and exclusion criteria for selection of literature.**

| Inclusion criteria | Exclusion criteria |
|---|---|
| Type of study: Studies (including preprints) on the preparedness and implementation of new and repurposed TB vaccines | Non-human trials and immunology studies |
| Country category: Low- and middle-income countries | TB vaccines trials on efficacy/ effectiveness, and safety |
| Age group: Adults and/or adolescents, with the specific age as defined in the study | Children (<9 years old), infants, neonates |
| Language: English only | BCG for neonates |
| Conducted between 2013–2023 | Books, guidelines, frameworks, roadmaps, presentations |

Studies focusing on vaccine candidates moving to or that are in phase 3 trials were included as were hypothetical vaccines. Hypothetical vaccines often align with the WHO Preferred Product Characteristics (PPC) [10] and do not adhere to the specific vaccine candidate characteristics currently in the pipeline. Literature covering other types of vaccines, (clinical/non-human) trials, and immunological studies were excluded. Studies on BCG vaccines (other than explicitly related to a revaccination strategy) were also excluded as were studies focused on the efficacy/effectiveness, and safety of new and repurposed TB vaccines. The primary focus of our review was on collecting primary data regarding the implementation of new and repurposed TB vaccines. Consequently, we chose not to include a grey literature search, as our expectation was that this type of literature might not predominantly contain primary data.

## Selection of articles

The literature assessment occurred in a sequential process. Initially, articles from PubMed and Medrxiv were imported into Zotero [13], where their abstracts and titles were evaluated for relevance. Simultaneously, PLOS journal articles were examined directly on its website, and those meeting the relevancy criteria were individually added to Zotero, unless already detected through the other searches. Subsequently, a comprehensive full-text screening was conducted to in- or exclude articles. In instances where articles reported secondary results, the initial paper was added. JSB conducted the initial selection, screening, and assessment of articles. Weekly meetings were held to discuss any articles for which JSB was uncertain about their inclusion in the review. These meetings further specified the inclusion process where needed. Additionally, PTP reviewed a random sample comprising 10% of the articles to ensure consistency in the inclusion process. The verification performed by PTP did not necessitate any changes to the selection of the articles. After title and abstract screening, JSB checked articles for their relevance through a full-text assessment. Duplicates were manually identified and removed during the abstract- and full-text screening. JSB consulted PTP to discuss articles for potential inclusion in cases of uncertainty. After full-text inclusion, the data were subtracted and collected in an extraction sheet using Excel 2023. An assessment (snowballing) of the references was performed on both the articles cited within the identified reviews and the final selected articles to ascertain any potentially relevant literature that was not captured within the initial search strategies. Similarly, any articles identified through the review of references and reviews followed the same structured inclusion process, as did any additional articles identified by the team.

## Data extraction and analysis

The Global Roadmap for Research and Development for TB Vaccines Roadmap and the ECVP [10, 11] provided key implementation factors for ensuring public health impact in shaping our analytical framework. Four pivotal implementation research components were evaluated on vaccine readiness: (1) Epidemiological impact (e.g. cases and deaths averted by vaccine), (2) Economic impact–on health systems, societies, macroeconomic-, value for money (e.g. cost per death or DALY averted, budget impact and impact on equity and social protection, including costing and cost-effectiveness), (3) Acceptability by the potential target population (i.e. the likely acceptability, including barriers and enablers, of the vaccine in the target populations and other key stakeholder groups), and (4) Implementation feasibility by potential target population (i.e. practicality of vaccine implementation, considering the logistics, delivery, and program related considerations). We adapted the conceptual framework from Atun [14] based on these key components (Fig 1).

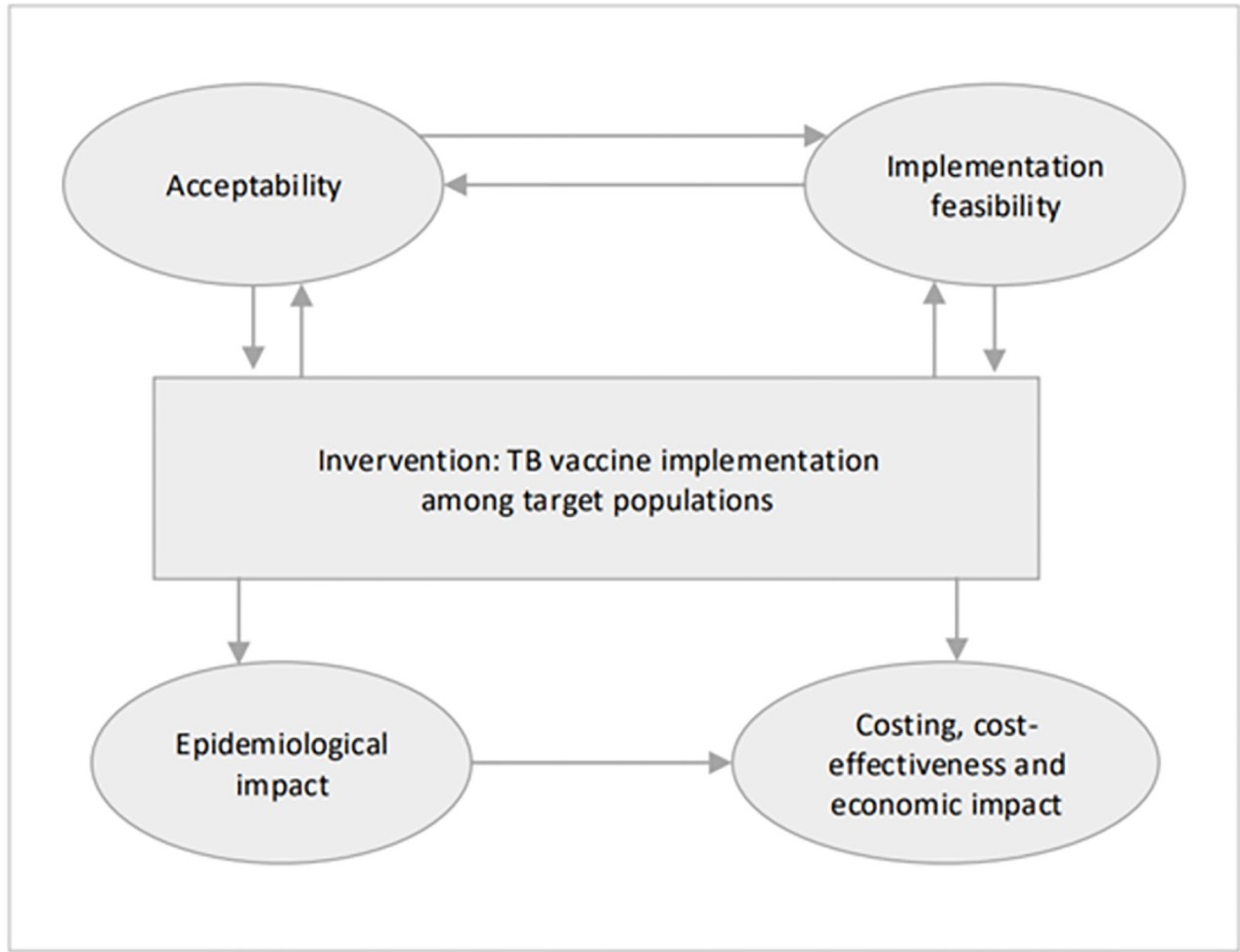

**Fig 1. Conceptual framework of factors influencing implementation of TB vaccination among adults and adolescents [11, 14].**

We listed the number of studies conducted within each component in an Excel sheet by country and evaluated each study based on the four vaccine readiness components. Results were reported for countries separately, outlining the vaccine characteristics (vaccine endpoint (PoD, PoI, prevention of recurrence (PoR)), duration of protection, VE), anticipated vaccination coverage, and time spans of the projected TB vaccine roll out. Implementation feasibility was further assessed through the lens of WHO's health systems building blocks [15]: health service delivery, health workforce, health information systems, accessibility, health financing, and leadership and governance.

## Results

### Articles output

We identified 1,807 articles of which 682 were from PubMed and 1,125 from Medrxiv (Fig 2). Following title and abstract screening of these initial articles and after removal of duplicates, 81 articles were included for full-text screening. The search strategy for PLOS journals resulted in the identification of two additional articles, which were included for full-text screening. Of

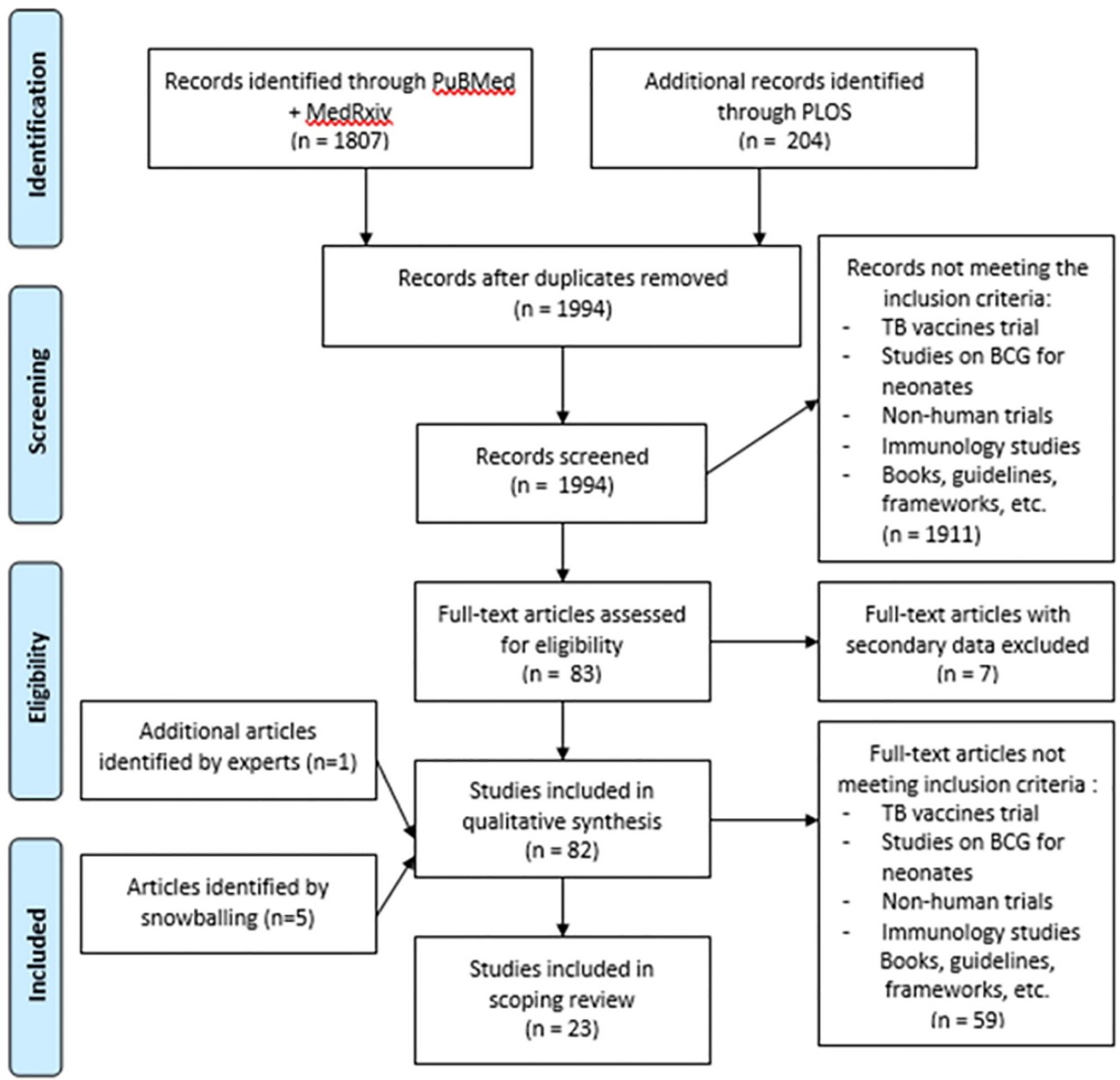

**Fig 2. PRISMA flow diagram [16] of article selection.**

the 83 articles for full-text inclusion that were reviewed, 66 articles did not meet the inclusion criteria of which seven articles referenced other articles with primary data. Five articles were identified through snowballing and one additional article was added by the team, resulting in 23 total articles for inclusion. Of the 23 articles, 18 were peer-reviewed, and 5 were scientific preprints.

## Study characteristics

A total of 23 articles were included in the review. S1 Table provides an overview of the number of included studies per country, the focus on vaccine readiness (epidemiological, costing/cost-

effectiveness/economic, acceptability, and implementation feasibility), vaccine candidate or profile, and vaccine endpoints (PoD, PoI, PoR). Most of the research was conducted in/ or focused on India (9 studies), followed by high burden LMICs in general (7 studies), and articles specific by country: South Africa (7 studies), China (6 studies), Indonesia (1 study), and Cambodia (1 study). Among the key research topics on TB vaccine implementation, epidemiological impact was the most frequently assessed topic (16 studies), followed by costing, cost-effectiveness, and/or economic impact (11 studies). One article addressed acceptability within the context of implementation feasibility, with a specific focus on exploring implementation strategies and health system readiness while five articles assessed implementation strategies. Various vaccine candidates and profiles were covered in the literature, including M72 /AS01e (8 studies), BCG revaccination (5 studies), and hypothetical vaccines (15 studies). Endpoints for the hypothetical vaccines, most frequently encompassed PoD (14 studies), while PoI (6 studies) was less common; no study modelled a PoR endpoint (see S2 Table).

## Epidemiological and costing, cost-effectiveness, and/or economic modelling studies

A total of 21 studies estimated the epidemiological and/or costing, cost-effectiveness, and/or economic benefits of new and repurposed TB vaccines as shown in Tables 2 and 3. The studies were conducted across multiple LMICs, with India (8 studies) leading, followed by South Africa, China, Indonesia, and Cambodia. Epidemiological impact projection took precedence, as 16 studies estimated the effects of vaccines on health outcomes. Additionally, 11 studies explored the costing, cost-effectiveness, and/or economic implications of vaccine implementation, emphasizing the comprehensive evaluation of benefits. The studies examined various vaccine candidates and profiles, with hypothetical vaccines being the most common (14 studies), followed by M72/AS01E (PoD) (7 studies) and BCG revaccination (PoI) (3 studies). The primary endpoints of the hypothetical vaccine profiles modelled with were PoD (13 studies) and PoI (5 studies) (see S3 and S4 Tables).

## Epidemiological impact

Epidemiological impact studies were mostly conducted for India (6) or for LMICs/various high burden countries (4), China (4), and South Africa (4). Vaccine candidates and profiles modelled with were hypothetical vaccines (11), M72/AS01E (5), and BCG revaccination (3). Of the studies modelling with a hypothetical vaccine, 9 studies used a PoD endpoint, 5 used PoI, and none used PoR. Most studies (11) focused on the general adult/adolescent population, 1 on adolescents only, 1 on the general population, 1 on miners, 1 on people with diabetes mellitus (DM), and one within a geographic hotspot with a high TB incidence (see S3 Table).

Overall, the epidemiological impact of a new and repurposed PoD TB vaccine was found to be substantial compared to a no TB vaccine scenario (beyond BCG vaccination in neonates), with the highest health gains projected within the groups/countries with the highest TB burden (see Table 2) [17–21]. Based on studies focusing on a selection of LMICs, the delivery strategy with the highest projected epidemiological impact consisted of routine vaccination with an instantly rolled out mass vaccination campaign, and concurrent with other public health interventions such as improved diagnostics, treatment, and reduced delay in health-seeking [17–21]. A new or repurposed PoD TB vaccine with an efficacy of 60% was found to help reach the End TB targets if a minimal coverage of 72% could be achieved in South Asia, Central Asia Republics and Europe (CAR/EU) and 40% in Sub-Saharan countries. This difference for Sub-Saharan countries is due to the projected impact of TPT uptake in PLHIV [17]. Additionally, a

**Table 2. Study outcomes from studies modelling health impacts.**

| Country | Author, year | TB[1] vaccine | Endpoint (VE[2]) | Duration of protection | Coverage | Timespan | Target population | Implementation | Projected impact |
|---|---|---|---|---|---|---|---|---|---|
| 105 LMICs[3] | Clark, 2023 [19] | Hypothetical vaccine[4] | PoD[5] (50%) | 10y | • Routine vaccination: 80% • Mass vaccination: 70% | 2028–2050 | Adults and adolescents | • Routine vaccination: 9years olds • Mass vaccination: 10 + year olds. | **Routine vaccination only:** 8.8 million (95% UR[6]: 7.6–10.1 million) TB cases could be averted. 1.1 million (95% UR: 0.9–1.2 million) TB deaths could be averted **Routine + one time mass vaccination** 44 million (95% UR: 37.2–51.6 million) TB cases could be averted. 5 million (95% UR: 4.6–5.4 million) TB deaths could be averted. **Quick scale up routine[7] + one time mass vaccination** 65.5 million (95% UR: 55.6–76.0 million) TB cases could be averted 7.9 million (95% UR: 7.3–8.5 million) TB deaths could be averted |
| 91 LMICs[3] | Knight, 2014 [21] | Hypothetical vaccine[4] | PoD[5] (60%) | 10y | • Routine vaccination: Coverage based on school attendance level. • Mass vaccination: 75% | 2024–2050 | Adults and adolescents | • Routine vaccination: 10-year-olds • Mass vaccination: 11 +year olds | 17 million (95% range: 11–24 million) TB cases could be averted |
| 30 high RR-TB burden countries[8] | Fu, 2021 [18] | M72/AS01E | PoD[5] (50%) | 10y | • Routine vaccination: Coverage based on school attendance level. • Mass vaccination: 72–76% | 2020–2035 | Adults and adolescents | • Routine vaccination: 15-year-olds • Mass vaccination: 15 + year olds | **Vaccination only** 620,000 (95% CrI[10]: 516,000–867,000) RR-TB[9] cases could be averted **Vaccination + improved management[13] of RR-TB** 831,000 (95% CrI: 643,000–1,170,000) RR-TB cases could be averted |
| 24 high burden countries[12] | Arinaminpathy, 2022 [17] | Hypothetical vaccine[4] | PoD[5] (60%) | 10y | • South Asia: 72% • Central Asian republics/Europe: 72% • Sub-Saharan Africa: 40% | 2022–2030 | Adults and adolescents | • Mass vaccination: scaled up linearly | **Mass vaccination + combination other interventions[11]** **South Asia** Annual incidence rate from 325 TB cases per 100,000 to <75 TB cases per 100,000 Annual mortality rate from 32 TB deaths per 100,000 to 7 TB deaths per 100,000 **Sub-Saharan Africa** Annual incidence rate from 205 TB cases per 100,000 to 75 TB cases per 100,000 Annual mortality rate from 47 TB deaths per 100,000 to 8 TB deaths per 100,000 **Central Asian republics/Europe** Annual incidence rate from 50 TB cases per 100,000 to 19 TB cases per 100,000 Annual mortality rate from 5 TB deaths per 100,000 to <2 TB deaths per 100,000 |
| China, India, South Africa | Harris, 2020 [24] | BCG revaccination & M72/AS01E | PoD[5] & PoI[7] (100%) | 10y | • Routine vaccination: 80% • Mass vaccination: 70% | 2025–2050 | Adults and adolescents | • Routine vaccination:9-year-olds • Mass vaccination: 10 + year olds | **Vaccine with PoI & PoD** **China** 11.6 million (UR[6]: 10.2 to 12.6 million) TB cases could be averted. 0.3 million (UR: 0.1 to 0.5 million) TB deaths could be averted. **South Africa** 4.3 million (UR: 2.5 to 7.0 million) TB cases could be averted. 0.9 million (0.5 to 1.6 million) TB deaths could be averted. **India** 4.3 million (UR: 2.5 to 7.0 million) TB cases could be averted. 0.9 million (UR: 0.5 to 1.6 million) TB deaths could be averted. |
| China, India | Weerasuriya, 2021 [28] | Hypothetical vaccine[4] | PoD[5] & PoI (50%) | 10y | • Routine vaccination: 80% • Mass vaccination: 70% | 2027–2050 | Adults and adolescents | • Routine vaccination: 9-year-olds • Mass vaccination: 10 + year olds | **Pre-infection and post-infection vaccine[15]** **India** 2.0 million (UI[16]: 1.4–4.1 million) RR/MDR-TB cases could be averted 0.4 million (UI: 0.3–0.7 million) RR/MDR-TB deaths could be averted **China** 2.1 million (UI: 1.1–2.7 million) RR/MDR-TB cases could be averted 0.1 (UI:0.0–0.2) million RR/MDR-TB deaths could be averted. **Post-infection vaccine[15]** **India** 1.3 million (UI: 0.9–2.6 million) RR/MDR-TB cases could be averted 0.3 million (UI: 0.2–0.4 million) RR/MDR-TB deaths could be averted **China** 0.7 million (UI:0.5–0.9 million) RR/MDR-TB cases could be averted 0.04 million (UI:0.02–0.06 million) RR/MDR-TB deaths could be averted |
| India, Indonesia | Silva, 2021 [27] | M72/AS01E | PoD[5] (49.7%) | lifelong | • Mass vaccination: 90% | 2020–2050 | Adults and adolescents | • Mass vaccination: coverage reached in 10 years | **India** 2.3 million TB deaths could be averted. **Indonesia** 695,000 TB deaths could be averted. |

(Continued)

**Table 2.** (Continued)

| Country | Author, year | TB[1] vaccine | Endpoint (VE[2]) | Duration of protection | Coverage | Timespan | Target population | Implementation | Projected impact |
|---|---|---|---|---|---|---|---|---|---|
| Cambodia | Renardy, 2019 [29] | Hypothetical vaccine[4] | PoD[5] & PoI (100%) | 10-15y | • 80-99% | 2021-2075 | Adults and adolescents | • Mass vaccination of varying age groups | TB cases could go from 300 per 100,000 to <50 per 100,000 depending on age groups. |
| China | Harris, 2019 [22] | Hypothetical vaccine[4] | PoD[5] (60%) | 10y | • 60% | 2025-2050 | Adolescents and elderly | • Routine vaccination: 15-year-olds<br>• Mass vaccination: 16-19 years olds<br>• Mass vaccination: 60 64-year-olds | **Pre-infection**[15]<br>**Adolescents**<br>248,000 (UI[16]: 214,000-292,000) TB cases could be averted<br>3,000 (UI: 2,000-5,000) TB deaths could be averted<br>**Elderly**<br>370,000 (UI: 287,000-504,000) TB cases could be averted<br>9 (UI: 5-21) TB deaths could be averted<br>**Post-infection latency only**[18]<br>**Adolescents**<br>8,000 (UI: 6,000-11,000) TB cases could be averted<br>90 (UI: 50-200) TB deaths could be averted<br>**Elderly**<br>658,000 (UI: 131,000-1081,000) TB cases could be averted<br>16,000 (UI: 2,000-45,000) TB deaths could be averted<br>**Post-infection latency or recovered**[18]<br>**Adolescents**<br>12,000 (UI: 9,000-16,000) TB cases could be averted<br>140 (UI: 70-290) TB deaths could be averted<br>**Elderly**<br>1,295,000 (UI: 1,037,000-1,469,000) TB cases could be averted<br>33,000 (UI: 16,000-65,000) TB deaths could be averted<br>**Pre-infection and post-infection**[15]<br>**Adolescents**<br>259,000 (UI: 224,000-304,000) TB cases could be averted<br>3,000 (UI: 2,000-5,000) TB deaths could be averted<br>**Elderly**<br>1,643,000 (UI: 1,403,000-1,893,000) TB cases could be averted<br>42,000 (UI: 22,000-81,000) TB deaths could be averted |
| | Liu, 2017 [30] | Hypothetical vaccine[4] | PoI[14] (75%) | lifelong | • 100% | 2018-2035 | General population | • Periodic vaccination of all ages (except neonatal which has its own campaign) | **Mixed vaccination**[19]<br>The proportion of infectious population reduces to <0.002 |
| India | Awad. 2020 [31] | Hypothetical vaccine[4] | PoD[5] & PoI[14] (50%) | conservative: 10y combination: lifelong | • 50% | 2020-2050 | People with diabetes mellitus | • Not specified | **Conservative**[20] **post-exposure vaccine**<br>1.7 million cases could be averted<br>**Campaign with pre- and post-exposure vaccine**<br>7.1 million TB cases could be averted |
| | Clark, 2023 [32] | M72/AS01E, BCG revaccination, hypothetical vaccine | PoD[5] & PoI[14] (50%) | 10y | • Routine vaccination: 80%<br>• Mass vaccination: 70% | 2025-2050 | Adults and adolescents | • Routine vaccination: 15-year-olds<br>• Mass vaccination:16 + year olds | **M72/AS01E**<br>12.7 (95% UI[16]: 11.0-14.6) million TB cases could be averted<br>2.0 (95% UI: 1.8-2.4) million TB deaths could be averted<br>**BCG revaccination**<br>9.0 (95% UI: 7.8-10.4) million TB cases could be averted<br>1.5 (95% UI: 1.3-1.8) million TB deaths could be averted |
| | Shresth, 2016 [26] | Hypothetical vaccine[4] | PoD[5] (60%) | 10y | • Routine vaccination: 80%<br>• Mass vaccination in hotspot[21]: 80%<br>• Mass vaccination general population 14-18% | 10 years | Adults and adolescents | • Routine vaccination:10-year-olds<br>• Mass vaccination of 20 +year olds in hotspot or general population | **Spatially targeted vaccination (STV) program**<br>TB incidence reduction of 28% in the state of Gujarat<br>**Universal targeted vaccination program**<br>TB incidence reduction of 24% in the state of Gujarat |
| South Africa | Dye, 2013 [33] | BCG revaccination | PoI[14] (80%) | 10y | 100% | Not provided | Adolescents | • Vaccination: 15-year-olds | 17% of cases could be averted |
| | Shresth, 2017 [25] | Hypothetical vaccine[4] | PoD[5] (60%) | 10y | • Not provided | 20 years | Miners (18-60-year-olds) and (mining) labor sending community | • Routine vaccination of new miners at recruitment<br>• Integration with medical examination | **Targeting miners**<br>8,090 (95% range[22], 3,750-13,300) TB cases could be averted<br>**Targeting (mining) labor sending community**<br>5,510 cases (95% range, 2,360-10,000) TB cases could be averted |

(Continued)

**Table 2.** (Continued)

| Country | Author, year | TB[1] vaccine | Endpoint (VE[2])[5] | Duration of protection | Coverage | Timespan | Target population | Implementation | Projected impact |
|---|---|---|---|---|---|---|---|---|---|
| | Jayawardana, 2022 [23] | M72/AS01E | PoD[5] (50%) | 5y | • Routine vaccination: 40%<br>• Mass vaccination: 60% | 2025–2050 | Adults and adolescents, PLHIV[23] | • Routine vaccination: 18–50-year-olds<br>• Mass vaccination: 18–50-year-olds | **All 18–50-year-olds**<br>**Mass + routine vaccination**<br>315,256 TB cases could be averted.<br>61,718 TB deaths could be averted.<br>**2 mass vaccination campaigns**<br>490,008 TB cases could be averted.<br>96,417 TB deaths could be averted.<br>**All PLHIV[23] adults**<br>**Mass+ routine vaccination**<br>209,524 TB cases could be averted.<br>42,143 TB deaths could be averted.<br>**2x mass vaccination**<br>367,862 TB cases could be averted.<br>73,191 TB deaths could be averted |

[1] . TB = tuberculosis,

[2] . VE = vaccine efficacy,

[3] . LMICs = Low-income countries (LIC), and lower and upper middle-income countries (LMIC). LICs are countries with a GNI per capita of $1,135 or less and LMICs with a GNI per capita between $1,136 and $4,465. Upper middle-income economies are those with a GNI per capita between $4,466 and $13,845. GNI per capita may differ per year (World Bank Country and Lending Groups – World Bank Data Help Desk),

[4] . Hypothetical vaccine = Hypothetical vaccines often align with the WHO Preferred Product Characteristics (PPC) [10] and do not adhere to the specific vaccine candidate characteristics currently in the pipeline,

[5] . PoD = prevention of disease,

[6] . UR = uncertainty range,

[7] . Quick scale up = In the accelerated scale-up scenarios, to more resemble the pace of COVID-19 vaccine introduction, all countries introduced vaccines in 2025 with coverage targets reached instantly,

[8] . High RR-TB high burden countries = India, Pakistan, Indonesia, Philippines, Myanmar, Bangladesh, Ethiopia, Russia Federation, Ukraine, south Africa, Mozambique, DR Congo, Zimbabwe, Nigeria, Thailand, Angola, Kenya, China, Vietnam, DPR Korea, Kazakhstan, Uzbekistan, Somalia, Peru, Kyrgyzstan, Papua New Guinea, Tajikistan, Belarus, Republic of Moldova, Azerbaijan,

[9] . RR-TB = Rifampicin resistant TB,

[10] . CrI = 95% Bayesian credible intervals,

[11] . combination other interventions = improved diagnostics, routine TB services, upstream case-finding, preventive therapy, mass vaccination,

[12] . High burden countries = Afghanistan, Bangladesh, Burma, Cambodia, India, Indonesia, Pakistan, Philippines, Vietnam, DR Congo, Ethiopia, Kenya, Malawi, Mozambique, Nigeria, South Africa, Tanzania, Uganda, Zambia, Zimbabwe, Kyrgyz Republic, Tajikistan, Ukraine, Uzbekistan,

[13] . Improved management = the detection of RR–TB at the point of TB diagnosis is increased to 85%, and second-line treatment success is increased to 75%, in countries that have not yet achieved these targets routine TB services, upstream case-finding, preventive therapy, mass vaccination,

[14] . PoI = prevention of infection,

[15] . pre-infection and post-infection vaccine = a vaccine effective in people regardless of infection status; pre–infection = vaccine is effective only in uninfected people; a post-infection vaccine is effective only in people who have been infected,

[16] . UI = uncertainty intervals,

[17] . MDR-TB = multidrug resistant TB,

[18] . post-infection latency only = a vaccine effective only in individuals with infection; post-infection latency or recovered = a vaccine effective in infected individuals or recovered,

[19] . mixed vaccination = routine vaccination and periodic vaccination,

[20] . conservative scenario = duration of protection is 10 years and no reduction in infectiousness of those with TB disease nor a reduction in progression rate to TB disease for those who are slow progressors,

[21] . hotspot = an area with a high TB burden,

[22] . Range = In this article, the 95% range represents variation across multiple model simulations, ie, the 2.5th and 97.5th percentile of the simulated result,

[23] . PLHIV = people living with HIV.

**Table 3. Main outcomes from studies modelling costs, cost-effectiveness, and economic impact.**

| Country | Author, year | TB[1] vaccine | Endpoint (VE[2]) | Duration of protection | Coverage | Time span | Target population | Implementation | Projected impact |
|---|---|---|---|---|---|---|---|---|---|
| 105 LMICs[3]* | Portnoy, 2023 [34] | Hypothetical vaccine[4] | PoD[5] (50%) | 10y | • Routine vaccination: 80% <br> • Mass vaccination: 70% | 2028–2050 | Adults and adolescents | • Routine vaccination: 9-year-olds <br> • Mass vaccination: 10 + year olds | **Incremental net monetary benefit for countries in which vaccination was cost effective.** $372 billion (95% credible intervals: $283-$474 billion) |
| | Portnoy, 2022 [35] | Hypothetical vaccine[4] | PoD[5] (50%) | 10y | • Routine vaccination: 80% <br> • Mass vaccination: 70% | 2028–2080 | Adults and adolescents | • Routine vaccination: 9-year-olds <br> • Mass vaccination: 9 + year olds | **Total absolute GDP[4] gains in LMICs** $1,618 billion (95% UI[7]: $764- $2,988 billion) |
| | Portnoy, 2022 [20] | Hypothetical vaccine[4] | PoD[5] (50%) | 10y | • Routine vaccination: 80% <br> • Mass vaccination: 70% | 2028–2050 | Adults and adolescents | • Routine vaccination: 9 years olds <br> • Mass vaccination: 10 + year olds. | **Total averted costs borne by TB-affected households.** $38.9 (95% credible intervals: $36.6–$41.5) billion could be averted **Averted number of households with catastrophic costs** $22.9 (95% credible intervals: 21·4–24·5) million households facing catastrophic costs could be averted |
| 96 LMICs[3]* | Knight, 2014 [21] | Hypothetical vaccine[4] | PoD[5] (80%) | 10y | • Routine vaccination: coverage based on school attendance level <br> • Mass vaccination: 75% | 2024–2050 | Adults and adolescents | • Routine vaccination: 10-year-olds <br> • Mass vaccination: 11 + year olds | **Cumulated reduction TB treatment costs** **LIC[3]** $5.3 million **LMIC[3]** $35.6 billion **UMIC[3]** $133.4 billion |
| China, India | Weerasuriya, 2021 [28] | Hypothetical vaccine[4] | PoD[5] & PoI[9] (50%) | 10y | • Routine vaccination: 80% <br> • Mass vaccination: 70% | 2027–2050 | Adults and adolescents | • Routine vaccination: 9-year-olds <br> • Mass vaccination: 10 + year olds | **Total vaccination program costs** **China** $41.5 ($39.8–$42.6) million **India** $38.6 ($37.1–$39.9) million **TB program savings** **China** $5.2 ($3.9–$6.8) million **India** $19.4 ($13.0–$27.2) million |

*(Continued)*

**Table 3.** (Continued)

| Country | Author, year | TB[1] vaccine | Endpoint (VE[2]) | Duration of protection | Coverage | Time span | Target population | Implementation | Projected impact |
|---------|-------------|---------------|------------------|------------------------|----------|-----------|-------------------|----------------|------------------|
| | Weerasuriya, 2021 [37] | M72/AS01E | PoD[5] (50%) | 10y | • 70% | 2027–2050 | Adults and adolescents | • Mass vaccination: 10 + year olds<br>• Mass vaccination: various age groups | **Country-specific willingness to pay threshold.**<br>**China**<br>$15 billion (UI: $12-$29) total cost at a willingness to pay of $3,650 per DALY[11] averted and $6 billion (UI: $4-$7) of total costs for vaccinating only the highest efficiency group (60-60y)<br>**India**<br>$21 billion (UI[7]: $16-$27) total cost at a willingness to pay of $264 per DALY averted and $5 billion (UI:$4-$6) of total costs for vaccinating only the highest efficiency group (50-59y) |
| India, Indonesia | Silva, 2021 [27] | M72/AS01E | PoD[5] (49.7%) | 10y | • 90% | 2020–2050 | Adults and adolescents | • Mass vaccination: coverage reached in 10 years | **Vaccine introduced in 2025.**<br>**India**<br>$2 trillion losses in income could be averted.<br>**Indonesia**<br>$680 billion losses in income could be averted |
| India, South Africa | Harris, 2022 [36] | M72/AS01E | PoD[5] (50%) | 15y | • Routine vaccination (10years old): 80%<br>• Routine vaccination (15years old): 80%<br>• Routine vaccination (18 years old): 50% | 2025–2050 | Adolescents | • Routine vaccination: 10-year-olds<br>• Routine vaccination: 15-year-olds<br>• Routine vaccination: 18-year-olds | **Pre- and post-infection efficacy[10]; 18-year-olds, 50% coverage**<br>**India**<br>Cost-saving with $-135.1 (-324.4, 19.6) cost per DALY[11] averted from a societal perspective<br>**South Africa**<br>Cost-effective with $290.1 (58.3, 723.1) cost per DALY[11] averted from a societal perspective |
| India | Clark, 2023 [32] | M72/AS01E, BCG revaccination, hypothetical vaccine | PoD[5] & PoI[9] (50%) | 10y | • Routine vaccination: 80%<br>• Mass vaccination: 70% | 2025–2050 | Adults and adolescents | • Routine vaccination: 15-year-olds<br>• Mass vaccination: 16 + year olds | **Annual incremental program cost**<br>**BCG revaccination**<br>US$23 million<br>**M72/AS01E**<br>$190 million<br>**ICER[12] ($/DALY[11] averted) from a societal perspective.**<br>**M72/AS01E**<br>$139 (77–229) per DALY[11] averted<br>**BCG revaccination**<br>$17 (cost-saving<71) per DALY[11] averted |

(*Continued*)

**Table 3.** (*Continued*)

| Country | Author, year | TB[1] vaccine | Endpoint (VE[2]) | Duration of protection | Coverage | Time span | Target population | Implementation | Projected impact |
|---|---|---|---|---|---|---|---|---|---|
| South Africa | Dye, 2013 [33] | BCG revaccination | PoI[9] (80%) | 10y | • 100% | - | Adolescents | • Vaccination of 15-year-olds | **Cost-effectiveness** Between $52 –$4,540 per DALY[11] averted depending on the vaccine price and VE[2] |
| | Jayawardana, 2022 [23] | M72/AS01E | PoD[5] (50%) | 5y | • Routine vaccination: 40% • Mass vaccination: 60% | 2025–2050 | Adults and adolescents, PLHIV | • Routine vaccination: 18–50-year-olds • Mass vaccination:18-50-year-olds | **Most cost-effective strategy: 2 mass vaccination campaigns for 18–50-year-olds** $184 million reduced TB related costs **Incremental cost of 2 mass vaccination campaigns for 18–50 year olds** $417 (IQR[13] 400–433 million) million |

[1].TB = tuberculosis,

[2].VE = vaccine efficacy,

[3]. LMICs = Low-income countries (LIC), and lower and upper middle-income countries (LMIC). LICs are countries with a GNI per capita of $1,135 or less and LMICs with a GNI per capita between $1,136 and $4,465. Upper middle-income economies are those with a GNI per capita between $4,466 and $13,845. GNI per capita may differ per year (World Bank Country and Lending Groups – World Bank Data Help Desk),

[4]. Hypothetical vaccine = Hypothetical vaccines often align with the WHO Preferred Product Characteristics (PPC) [10] and do not adhere to the specific vaccine candidate characteristics currently in the pipeline,

[5].PoD = prevention of disease,

[6].GDP = gross domestic product,

[7].UI: uncertainty intervals,

[8].catastrophic costs = instances where the patient costs of TB disease—the sum of direct medical costs, direct non-medical costs, and indirect costs—exceeded 20% of total annual income for the TB-affected household,

[9].PoI = prevention of infection,

[10].pre- and post-infection efficacy = effective irrespective of whether the recipient is infected with M. tuberculosis at the time of vaccination,

[11].DALY = disability-adjusted life years,

[12].ICER = incremental cost-effectiveness ratio,

[13].IQR = interquartile range.

scenario with a vaccine with a longer duration of protection, would avert more cases and require a lower number needed to vaccinate (NNV).

Country-specific studies revealed differences in impact as vaccine characteristics, target population and implementation strategy, timing of roll out, TB burden, and vaccination coverage rates vary. In China, a PoD vaccine targeting older people, protective in both *mycobacterium* tuberculosis (Mtb.) infected and recovered from TB populations, could deliver the highest impact [22]. In South Africa, Jayawardana et al. [23] projected the largest epidemiological impact with a strategy implementing a sequence of two mass vaccination campaigns and a second South African study found that a PoD vaccine could be most impactful if effective in both uninfected and infected people, and safe for use among PLHIV [24]. Moreover, Shrestha et al. [25] found a greater impact of a miner-targeted vaccine strategy compared to a community-targeted strategy in associated labour sending communities in South Africa. This was due to the higher proportion of adult men in the miner target population. In India, Silva et al. found that a vaccination program's effectiveness targeting high incidence geographic areas depends on interaction of populations between geographic areas. Vaccination of people in

ahigh TB incidence area with high population density and movement, could reduce incidence 1.6-fold or more [26]. Another study showed in Indonesia and India that vaccinating either all ages or 18–49 year olds could bring substantial health and economic gains, and both health and economic gains would be more substantial with vaccinating all age groups [27].

## Costing, cost-effectiveness, and/or economic impact

Eleven studies investigated costing, cost-effectiveness, and economic impacts (see Table 3 and S4 Table). The distribution of studies included general LMICs (4) and studies in India (5), South Africa (3), China (2), and Indonesia (1). Across the studied countries, the M72/AS01E vaccine candidate was most frequently studied in India (4 studies), whereas hypothetical PoD vaccines were investigated across India (5), South Africa (3), China (2), and Indonesia (1). Costing, cost-effectiveness, and economic measurement outcomes differed across studies, with cost-effectiveness estimated for all except one country (Indonesia). Overall, adults and adolescents from the general population were the most frequently modelled target populations.

In studies focused on LMICs, adult/adolescent TB vaccine strategies could be economically beneficial compared to situations without new and repurposed TB vaccines. However, this is dependent on the TB burden, prevalence of HIV co-infection, vaccine program approaches, and ultimately the efficacy and duration of protection of the vaccine [20, 21, 34, 35]. An adult/adolescent TB vaccine was found to be cost-effective in most countries and cost-saving in 58 LMICs [34], with higher TB incidence countries gaining the most benefit. Furthermore, an adult/adolescent vaccine program could reduce catastrophic costs by 40%, with the highest gains in the lowest income quintiles [20].

Country-specific estimates also illustrate that costs, cost-effectiveness, and economic impact is dependent on vaccine characteristics, TB burden, country-specific context and thresholds for cost-effectiveness, coverage, and timing of a program. In India, M72/AS01E is 7-times more cost-effective than BCG revaccination despite cost-effectiveness being evident across all scenarios [32]. Additionally, Harris et al. found that only a vaccine which includes efficacy in post- & pre-infected people was cost-effective in India because, the TB epidemic is equally driven by new infections of susceptible people as re-infection of people with TB infection (TBI)and due to their level of thresholds for cost-effectiveness [28, 36]. In South Africa, a PoD vaccine could be highly cost-effective and even the least effective PoD vaccine scenario (80% coverage of 10 year olds, post-infection protection) remained below the cost-effectiveness threshold [36]. The inclusion of HIV-related costs was a relevant contextual factor in South Africa as mortality caused by TB among PLHIV will be prevented by the vaccine. In China, Weerasuriya et al. found that a PoD vaccine effective in both TB infected and uninfected people could be most cost-effective based on rifampicin-resistant and multidrug-resistant tuberculosis (RR/MDR-TB) reductions [28]. Another study by Harris et al. found that in India and South Africa, the costs of vaccinating 18 year olds with 50% coverage was equal to the costs of vaccinating 15 year olds with 80% coverage, and had a similar impact [36]. Lastly, Silva et al. demonstrated that for India and Indonesia, a delay in implementation from 2020 to 2025 and the subsequent loss in economic impact could be recovered by vaccinating all ages instead of only a specific age group (18–49 year olds) [27].

## Acceptability

One study assessed the acceptability of TB vaccine implementation in India, China, and South Africa [38]. This article evaluated acceptability among high level vaccine policy and TB experts (Table 4). In all three countries, barriers included TB-related stigma affecting coverage during a mass vaccination campaign as well as affordability. Barriers to uptake specific to China

**Table 4. Factors influencing acceptability and demand of a new and repurposed TB vaccine [38].**

| | India | China | South Africa |
|---|---|---|---|
| **Beliefs and attitudes** | - | - | - |
| **Barriers** | • Stigma<br>• Negative publicity of side effects<br>• High cost<br>• Implementing the BCG revaccination | • Stigma<br>• Vaccine hesitancy<br>• High cost<br>• Low perceived level of protection | • Stigma<br>• Vaccine hesitancy<br>• The need for a second dose<br>• High cost |
| **Enablers /Facilitators** | • High awareness, especially among elderly<br>• Low price | • Routine vaccination<br>• Vaccination campaign Integrated within national immunization program.<br>• High awareness/health education<br>• Community engagement<br>• Low price<br>• Familiarity with the BCG vaccine if BCG revaccination is implemented | • Routine vaccination<br>• Strong political commitment<br>• Lessons learned from the COVID-19 vaccination program.<br>• Civil society engagement<br>• Low price<br>• Familiarity with the BCG vaccine if BCG revaccination is implemented |

included vaccine hesitancy and perceived level of protection. In India, experts perceived negative publicity of side effects as barrier to acceptability by the public and were hesitant about BCG revaccination resulting from previous trials showing no effects among Indian children. South African specific barriers included vaccine hesitancy and the need for a second dose of a vaccine candidate. The facilitators identified across countries included low price per dose, and public awareness of TB disease and the new and repurposed TB vaccine. Moreover, routine vaccination strategies, engaging communities/civil societies, and familiarity with the vaccine (in the case of BCG revaccination) were considered facilitators in the Chinese and South African contexts.

## Implementation feasibility

Five articles contained information on key considerations for implementation feasibility, including implementation strategies of an adult/adolescent TB vaccine in LMIC, and health system readiness [23, 25, 26, 38, 39]. The S5 Table shows the number of implementation feasibility studies by country. Three countries were investigated, with South Africa being the most represented.

## Implementation strategies by target population

All five articles contained information related to country-specific implementation strategies, in the context of China, India, and South Africa (S5 Table) [23, 25, 26, 38, 39]. Various strategies for TB vaccine implementation were identified, which are summarized in Table 5. Target populations that were similar across the countries were adolescents, adults, socially vulnerable groups, PLHIV and other people with immunocompromised systems, and healthcare workers [38, 39]. The elderly and high risk contacts were target populations evaluated in India and China [38] and in South Africa the mining-community was evaluated as a target population [25]. In the Indian context, a focus on high-risk populations by targeting high incidence geographic 'hotspots' was also assessed [26].

**Table 5. Overview of studies assessing implementation strategies.**

| Author, year | Country | TB[1] vaccine | Endpoint | Time span | Target population | Implementation strategies |
|---|---|---|---|---|---|---|
| Pelzer, 2022 [38] | China, India, South Africa | M72/AS01E &BCG revaccination | PoI[2]/ PoD[2] | 2025–2030 | • Adolescents<br>• Adults<br>• Elderly<br>• Socially vulnerable groups<br>• PLHIV<br>• Immunocompromised<br>• HCW<br>• Other | • Routine vaccinations<br>• Targeted vaccination programs<br>• Mass campaigns<br>• Vaccination campaign integrated into the existing healthcare systems |
| Hatherill, 2016 [39] | South Africa | Hypothetical vaccine[3] & BCG revaccination | PoI/PoD | Not specified | • Healthcare workers | • Vaccination campaign integrated within the occupational health infrastructure |
| Shresta, 2016 [26] | India | Hypothetical vaccine[3] | PoD | Not specified | • Population living in geographic high incidence TB 'hotspots'[4] | • Spatially targeted vaccine program |
| Shresta, 2017 [25] | South Africa | Hypothetical vaccine[3] | PoD | 20 years | • Mining population | • Routine vaccination of miners at recruitment<br>• Vaccination integrated with the medical examination |
| Jayawardana, 2022 [23] | South Africa | M72/AS01E | PoD | 2025–2050 | • Adults/adolescents and PLHIV[5] adults | • Mass vaccination combined with a routine annual vaccination of 18-year-olds<br>• two mass campaigns |

[1].TB = tuberculosis,

[2].PoI = prevention of infection; PoD = prevention of disease,

[3]. Hypothetical vaccine = Hypothetical vaccines often align with the WHO Preferred Product Characteristics (PPC) [10] and do not adhere to the specific vaccine candidate characteristics currently in the pipeline,

[4]. Hotspots = an area with a high TB burden,

[5]. PLHIV = people living with HIV.

## Health system readiness

Two articles discussed health system readiness for new and repurposed TB vaccine implementation in India, China, and South Africa (Table 6. Overview of studies assessing health system readiness) [38, 39]. For all three countries, there was a described need for an extensive monitoring and surveillance system to track potential adverse events. In India, it was mentioned that a mass vaccination campaign could be hindered by new of cold chain requirements, while routine vaccination could be more easily integrated within the existing public health system. Additionally, testing for Mtb. infection was considered too expensive and unfeasible in India. In South Africa, testing for Mtb. infection and the high prevalence of HIV infection -which may limit which vaccines could be safely used- were also potential challenges. In China, storage capacity was an additional factor that could delay effective implementation, yet the current procurement and supply chain could manage the integration of a new and repurposed vaccine program. In South Africa and China, insufficient staff capacity could hinder any future vaccination campaign.

## Discussion

In this scoping review, we found that there is limited body of evidence on implementation potential of new and repurposed TB vaccines among adults and adolescents. Most studies

**Table 6. Overview of studies assessing health system readiness.**

| Health system readiness | | India | China | South Africa[2] |
|---|---|---|---|---|
| Health service delivery | Barrier | • New cold chain requirements<br>• Testing for Mtb. infection[1] | • There may be limited storage capacity<br>• Storage at -80C | • Scarcity and price of the adjuvants (for M72/AS01E)<br>• Testing for Mtb. infection[1] and high prevalence of HIV infection. |
| | Facilitator | • Routine vaccination | • Current procurement and supply chain management systems can handle out roll. | • Single dosing administered by the lowest level healthcare worker without reliance on a supply chain.<br>• Information about the epidemiology, TBI3 and subnational BCG coverage |
| Health workforce | Barrier | - | • Staff workload | • Limited number of nurses |
| | Facilitator | - | - | - |
| HIS accessibility | Barrier | • Training of staff on monitoring and surveillance system for tracking potential adverse events | • Strengthen monitoring and surveillance system for tracking potential adverse events | • Strengthen of monitoring and surveillance system for tracking potential adverse events |
| | Facilitator | - | - | - |
| Health financing | | See Table 3 for the projected costs, cost-effectiveness, and economic impacts | See Table 3 for the projected costs, cost-effectiveness, and economic impacts | See Table 3 for the costs, cost-effectiveness, and economic impacts |
| Leadership | | - | - | - |
| Governance | | - | - | - |

[1].Mtb. infection = *mycobacterium* tuberculosis,

[2]. In South Africa, these components would also have to be in place in the hospital when targeting healthcare workers,

[3].TBI = tuberculosis infection.

assessed epidemiological or costing/cost-effectiveness and/or economic impacts of new and/or repurposed TB vaccines. Limited research looked at implementation feasibility and only one study contained information regarding acceptability of new and repurposed TB vaccines among high level stakeholders. The Sub-Saharan African region was underrepresented in all the studies assessed in this review. While our findings clearly highlight the availability of evidence that points toward potential impacts of new and repurposed TB vaccines among adults and adolescents, it also underscores the need for broader geographic and target population representation in the research field.

While the epidemiological studies showed that new and repurposed TB vaccines may avert millions of TB cases and TB deaths [19], the estimated impacts differed per country and were dependent on vaccine characteristics, target population and implementation strategy, the timing of roll out, the local TB burden, and vaccination coverage [22–27]. Costing, cost-effectiveness, and economic studies showed that in a majority of countries, and remarkably, in 58 LMICs, an adult/adolescent vaccine program could potentially not only be cost-effective but even yield cost-savings outcomes [34]. Vaccine characteristics, TB burden, cost-effectiveness, coverage, and timing all influence costing, cost-effectiveness, and economic-related outcomes [19–21, 27, 28, 32, 34, 36]. Vaccine candidates and profiles modelled with included hypothetical vaccines, M72/AS01E, and BCG revaccination like candidates. Vaccine efficacies ranged from 50%-100%, with coverage rates between 40%-100%, and duration of protection between 1 year and lifelong (see Tables 2 and 3). Most studies focused on PoD, some on PoI, and the target population was mainly general adults and adolescents. The study investigating factors influencing acceptability of new and repurposed TB vaccines in India, South Africa, and China showed that countries may differ in their preference for a vaccine candidate and barriers/

benefits. In all three countries, common barriers included TB-related stigma affecting coverage during a mass vaccination campaign as well as affordability. Facilitators identified across countries were low price per dose, and public awareness of TB disease and the new and repurposed TB vaccine [38]. Lastly, studies focusing on implementation feasibility showed that strategies could include mass campaign(s), integrated within existing routine vaccine programs, and targeted campaigns specifically for -for example- miners, elderly, PLHIV, and people with immunocompromising conditions. This will be dependent on country specific context in terms of burden and health system readiness and capabilities [23, 25, 26, 38, 39].

While this scoping review identified limited information on implementation feasibility, prior research regarding the implementation of other adult vaccines confirms the few findings and underscores the need for more research to ensure effective implementation, once a new and/or repurposed TB vaccine is available. Vaccine hesitancy, lack of availability of monitoring adverse events, affordability and funding/cost of delivery, and storage requirements were also challenges identified in previous studies focusing on other adult vaccines [40–42]. Additionally, need for estimating target population size, slow roll-out of the vaccination campaign, accessibility, and wars/conflicts were additional challenges mentioned in reviews specifically focusing on LMICs [42, 43]. A holistic and timely implementation planning process, expanding access, reducing out-of-pocket costs, one-dose vaccine programs, extensive community involvement, and implementation of programs in schools could circumvent some of these challenges [40, 42, 43]. Notably, as assessed in the review by Wang et al., introducing a new vaccine may also have broader positive effects on the general health system [44]. They found that besides reductions in the disease burden, improvements in disease and adverse events surveillance, training, cold chain and logistics capacity and injection safety were documented as beneficial effects. However, they also emphasize that opportunities for strengthening the broader health system were consistently missed during out roll [44]. Research in LMICs other than India, China, and South Africa was lacking, which is especially important due to three aspects. Firstly, most TB cases and deaths are concentrated in LMICs with eight countries comprising of around two-thirds of the burden worldwide [45]. Secondly, epidemiological variations among countries influence TB vaccine implementation. Therefore, factors such as TB/HIV prevalence, tuberculosis infections (TBI) rates, BCG vaccine coverage, and multi-drug resistant (MDR)-TB burden must be considered. Thirdly, while BCG revaccination and M72/AS01E vaccine candidates are advanced, they may not be suitable for all contexts and populations if proven effective in regulatory trials. While several vaccines (M72/AS01e and MTBVAC) are entering late stage trials, there is still a need for proven effectiveness in regulatory trials. Previous studies found no effect of the BCG revaccination against the development of TB infection or disease [46–48]. Therefore, investments in and development of new TB vaccine candidates will remain crucial.

Further clinical research outcomes of vaccine candidates in the pipeline will guide the parameters to consider in future modelling. Currently, research primarily relies on expert opinions, and it lacks essential acceptability assessments among end users. Although cross-country representation is significant, we need to consider specific target populations within each country. As shown in this review, different countries have varying target populations, each with distinct estimated impacts and coverage levels. It is vital to expand research to include other LMICs, various target populations, and other late-stage vaccine candidates. This broader approach will enhance our understanding of vaccine characteristics, leading to better decision-making and equitable distribution.

Understanding the TB burden and TB infection rates within various target populations across countries is paramount to optimizing the potential benefits of new and/or repurposed TB vaccines. This foundational knowledge serves as the basis upon which tailored vaccine implementation strategies can be formulated. Crucially, the development of implementation

strategies should be guided by feasibility and equity considerations. These strategies should be linked to the TB burden in specific regions, considering contextual factors and health disparities that impact the effective deployment of TB vaccines. The experiences of China, India, and South Africa, which have chosen routine vaccination as a practical implementation strategy, underscore the feasibility of integrating new and repurposed vaccines into existing vaccination programs. Consequently, we recommend future TB vaccine preparedness research to investigate country-specific TB/TBI rates within target populations.

Further examination of factors such as stigma, accessibility, availability, vaccine hesitancy, and vaccine confidence, along with an exploration of the mechanisms, facilitators, and barriers influencing these aspects, holds pivotal significance. These determinants exert a direct impact on the acceptance, adoption, and uptake of novel or repurposed TB vaccines. Considerations like community beliefs and attitudes, equitable representation, and individual choices regarding vaccination are essential for informing the landscape of acceptability and feasibility. Regrettably, the existing body of research exhibits a large gap in knowledge, as no prior studies have delved into these critical topics. Gallagher et al. [41] and McClure et al. [49] underscore the significance of addressing hesitancy and rumors, as these factors can detrimentally affect vaccine programs on a broader scale. The absence of research in this domain underscores the urgent necessity for comprehensive investigations and leveraging insights from existing research on non-TB vaccines aimed at better comprehending the dynamics that shape individuals' attitudes, perceptions, and behaviors towards a potential new TB vaccine.

Furthermore, it is of utmost importance to assess the readiness of healthcare systems in LMICs for the seamless integration of new and repurposed TB vaccines. Experience from the implementation research of COVID-19 vaccination has highlighted the critical importance of health system readiness [50]. This readiness is essential to mitigate delays and prevent losses in the implementation of TB vaccines and should encompass an evaluation of healthcare infrastructure, capacity, and potential obstacles, enabling the development of well-informed strategies that facilitate successful implementation. See S6 Table for an overview of the prioritized gaps for future research to guide new and repurposed TB vaccine implementation.

Our study has limitations. Notably, we conducted a scoping review, which is distinct from a comprehensive systematic review and we solely included articles written in English. While we employed diverse methods to comprehensively map the existing literature and identify gaps, it is possible that some studies meeting the inclusion criteria have been omitted. Nevertheless, our primary aim was to present a comprehensive overview of the current state of the field, rather than striving for an exhaustive compilation. We encourage future reviews to include non-English articles and consider conducting systematic review [51] depending on the scope and objective. Additionally, our scope was exclusively focused on TB vaccine implementation. Over recent years, however, much experience has been gained with implementation of vaccinations for adults and adolescents e.g., COVID-19, hepatitis B, and HPV. These lessons learned would be very relevant and valuable to the implementation of a new or repurposed TB vaccine in the same age groups. Despite these limitations, it is the first review to comprehensively map the existing literature on the implementation and impacts of new and repurposed TB vaccines. This provides stakeholders with a clear overview of the current body of knowledge, enhancing the understanding of stakeholders regarding the current landscape and aiding them in prioritizing future research in this area.

## Conclusion

With several TB vaccine candidates in the pipeline, there is an urgent need for research on the global implementation of effective TB vaccines among adult and adolescent populations,

including assessment of economic and epidemiological impacts in LMICs. Country-specific evidence on TB vaccine implementation is also imperative but is currently lacking. Addressing these gaps will be needed to prepare countries for novel TB vaccine implementation, upon approval, to ensure uptake through the design of optimal and equitable vaccine implementation strategies. Future studies should focus on generating evidence on understudied areas such as key factors influencing acceptability and implementation feasibility. Furthermore, geographic and population diversity should be seriously considered when planning further studies to understand TB vaccine implementation strategies.

## Supporting information

**S1 Text. Search strings.**
(DOCX)

**S1 Table. Number of included studies per country, vaccine readiness focus, vaccine candidate and profile, and vaccine endpoint.**
(DOCX)

**S2 Table. Overview characteristics included articles.**
(DOCX)

**S3 Table. Study characteristics and main measurement outcomes from epidemiological impact studies by country.**
(DOCX)

**S4 Table. Study characteristics and main measurement outcomes from studies modelling costing, cost-effectiveness, economic impacts by country.**
(DOCX)

**S5 Table. Characteristics of studies evaluating facets related to implementation feasibility by country.**
(DOCX)

**S6 Table. Prioritized gaps for future research to guide new and repurposed TB vaccine implementation.**
(DOCX)

## Acknowledgments

**Disclaimer:** This study/report/publication is made possible by the support of the American people through the United States Agency for International Development (USAID). The contents are the sole responsibility of (authors) and do not necessarily reflect the views of USAID, the United States Government, or consortium collaborators or members.

## Author Contributions

**Conceptualization:** Joeri S. Buis, Puck T. Pelzer.

**Data curation:** Joeri S. Buis.

**Formal analysis:** Joeri S. Buis.

**Methodology:** Joeri S. Buis, Puck T. Pelzer.

**Software:** Joeri S. Buis.

**Supervision:** Degu Jerene, Agnes Gebhard, Puck T. Pelzer.

**Validation:** Agnes Gebhard.

**Visualization:** Joeri S. Buis, Puck T. Pelzer.

**Writing – original draft:** Joeri S. Buis, Puck T. Pelzer.

**Writing – review & editing:** Joeri S. Buis, Degu Jerene, Agnes Gebhard, Roel Bakker, Arman Majidulla, Andrew D. Kerkhoff, Rupali J. Limaye, Puck T. Pelzer.

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
