## [Decision Letter · Decision Letter 0]

18 Mar 2024

PGPH-D-24-00052

Mapping the Existing Body of Knowledge on New and Repurposed TB Vaccine Implementation: A Scoping Review

Dear Dr. Joeri Sumina Sumina Buis 

Thank you for submitting your manuscript to PLOS Global Public Health. After careful consideration, we feel that it has merit but does not fully meet PLOS Global Public Health’s publication criteria as it currently stands. Therefore, we invite you to submit a revised version of the manuscript that addresses the points raised during the review process.

Both reviewers considered the manuscript acceptable but there are important considerations such as the scope of the literature search being limited only to English language publications; the methods used for screening and selecting articles for inclusion; and a more detailed description of studies involving BCG re-vaccination. 

We look forward to receiving your revised manuscript.

Kind regards,

Indira Govender, MBChB, MMed, FCPHM

Academic Editor

Journal Requirements:

1. We ask that a manuscript source file is provided at Revision. Please upload your manuscript file as a .doc, .docx, .rtf or .tex.

Additional Editor Comments (if provided):

Reviewers' comments:

Reviewer's Responses to Questions

**Comments to the Author**

1. Does this manuscript meet PLOS Global Public Health’s publication criteria? Is the manuscript technically sound, and do the data support the conclusions? The manuscript must describe methodologically and ethically rigorous research with conclusions that are appropriately drawn based on the data presented.

Reviewer #1: Partly

Reviewer #2: Yes

2. Has the statistical analysis been performed appropriately and rigorously?

Reviewer #1: N/A

Reviewer #2: I don't know

3. Have the authors made all data underlying the findings in their manuscript fully available (please refer to the Data Availability Statement at the start of the manuscript PDF file)?

Reviewer #1: Yes

Reviewer #2: Yes

4. Is the manuscript presented in an intelligible fashion and written in standard English?

Reviewer #1: Yes

Reviewer #2: Yes

5. Review Comments to the Author

Reviewer #1: Buis et al.

This is a useful overview of the currently available data on implementation of new and repurposed TB vaccines along four lines (health impact, economics, acceptability and feasibility) that together define the introduction and scalability of such vaccines in low- and middle income countries.

MAJOR COMMENTS

I am a bit concerned about the language restriction. The finding that limited (published) data is available may well reflect that no articles in Spanish (Latin America), French (francophone Africa), Portuguese (Brazil), Russian (GAMTBVac is in phase 3) and Chinese (M. vaccae, in fact the one novel TB vaccine that is being implemented in health practice) were included. This is in particular important for studies looking at feasibility and acceptability. I would urge the authors to at least screen other language articles with English abstracts.

Contrary to established practice there was no duplicate screening and article selection. Only a random sample was checked by the senior author. It would be important to know what proportion that sample was, and what exactly was checked. Only the abstracts, full texts?

There is quite some attention to BCG revaccination, referring to a single trial with a prevention-of-infection endpoint. However, there have been at least two randomized trials that showed no effect of BCG revaccination on prevention of TB disease. This requires explanation in the introduction section and reflection in the discussion section.

MINOR COMMENTS (LINES INDICATED):

Tables don't seem to number through.

Abstract

41 adult

Introduction

91 Reference 3 is unpublished and not accessible. Please provide alternative(s).

98-99 These include both vaccines for preventive indication and vaccines for therapeutic indication (MIP). It would seem important to make that distinction. In addition: there is no mention of M. vaccae.

107 Please define "repurposed".

114 Please explain why excluding TB infection is relevant.

Methods

150 Was there a protocol? Is that available? Was the review registered?

168 Who are meant by "the internal team"?

176 "moving to or that are"

177 Important to define "generic". Consider calling it "hypothetical" instead.

189 Provide reference for Zotero.

197 "thorough" - please clarify.

204 AIGHD/EDCTP - is that the same as the roadmap mentioned earlier?

208 missing ")"

211 "the potential"

Was any quality appraisal of the included articles conducted?

Results

240 Please explain "readiness focus" and "vaccine endpoints".

265 I don't think these studies were done in these countries but for these countries.

285 "Mycobacterium" (italicized); recovered from...?

291 "proportion" instead of "concentration"

294 what is meant by "well-connected"?

315 "cost" instead of "costing"

319 please clarify "due to the epidemic"

325 please clarify "RR-TB"

315-327 This observation is difficult to interpret unless the impact of the two approaches was similar. Was it?

332 Leave out table title.

348 part of sentence is missing

352 "All five"

365 Leave out table title.

368 What are "repurposed cold chain requirements"?

Discussion

379 "potential"

381 "followed by" - stylistically awkward sentence

408-409 please clarify "prior research confirms the few findings"

410-411 "monitoring and surveillance systems" - of what?

412 "need for estimating"

427-428 Please add this is only relevant if these vaccines are proved effective in regulatory trials.

430 "approvable" - by whom?

432 "opinion"; why "but"?

433 please clarify "necessary"

435-436 "other (...) candidates"

439 "paramount to"

453 why "subsequently"?

454 please clarify "conspicuous"

457-459 But there is an extensive literature on this topic for non-TB vaccines.

461 leave out "("

470 why "inadvertently"? It suggests the screening was not done too thoroughly.

474 "hepatitis"

Table 1

Children is defined up to what age (to separate them from adolescents)

Figure 2

Provide reasons for exclusion (grouped)

Table 4

Is there any information on the duration of vaccine protection assumed in these studies?

What is the rationale for the order of articles? It seems neither alphabetical not chronological.

Please provide clarification of all abbreviations.

Please be clear and consistent about what the numerical intervals represent. What does "range" mean in this regard?

Please clarify what is meant by "POI+POD", "pre-infection and post-infection"

Fu 2014: "interval"

Shrestha 2017: what are "repurposed miners"?

Harris 2019: what do the intervals refer to?

Liu 2017: please clarify "pulse"

Awad 2020: please clarify "conservative"

Shrestha 2016: is this incidence reduction population-wide or in the hotspots only?

Renardy 2019: "Yearly..." suggests that every year the same reduction is predicted. I don't think that is what is meant.

Table 5

See general comments Table 4

Weerasuriya 2021: WTP thresholds are generally for cost/DALY. Here they seem to refer to a total amount. Please clarify.

Harris 2022: Please provide an interpretation of the negative cost/DALY estimates.

Due 2013: What does the range of cost/DALY represent?

Jayawardana 2022: Please clarify "total incremental cost"

Table 10

See general comments Table 4

I assume "not specific" should read "not specified"

Shrestha 2017: if not specified, why is there a timespan?

Table 11

See general comments Table 4

HIS acceptability: I find this barrier surprising. All three countries have established pharmacovigilance systems. So, what exactly is meant here?

References

37 is duplicated as 42

Reviewer #2: This is a very well written manuscript addressing the growing concern that when we do find a new TB vaccine the public and governments will not accept it or add to the national programs. It is very important that these challenges be addressed timeouts to avoid further delay in stopping TB

6. PLOS authors have the option to publish the peer review history of their article (what does this mean?). If published, this will include your full peer review and any attached files.

**Do you want your identity to be public for this peer review?** For information about this choice, including consent withdrawal, please see our Privacy Policy.

Reviewer #1: **Yes: **Frank Cobelens

Reviewer #2: No

---

## [Decision Letter · Decision Letter 1]

19 Jun 2024

PGPH-D-24-00052R1

Mapping the Existing Body of Knowledge on New and Repurposed TB Vaccine Implementation: A Scoping Review

Dear Dr. Joeri Sumina Sumina Buis,

Thank you for submitting your manuscript to PLOS Global Public Health and for addressing the concerns raised by the reviewers. After careful consideration, we feel that it has merit but does not fully meet PLOS Global Public Health’s publication criteria as it currently stands. Therefore, we invite you to submit a revised version of the manuscript that addresses the points raised during the review process.

We look forward to receiving your revised manuscript.

Kind regards,

Indira Govender,

Academic Editor

Journal Requirements:

Additional Editor Comments (if provided):

Reviewers' comments:

Reviewer's Responses to Questions

**Comments to the Author**

1. If the authors have adequately addressed your comments raised in a previous round of review and you feel that this manuscript is now acceptable for publication, you may indicate that here to bypass the “Comments to the Author” section, enter your conflict of interest statement in the “Confidential to Editor” section, and submit your "Accept" recommendation.

Reviewer #1: (No Response)

Reviewer #2: All comments have been addressed

2. Does this manuscript meet PLOS Global Public Health’s publication criteria? Is the manuscript technically sound, and do the data support the conclusions? The manuscript must describe methodologically and ethically rigorous research with conclusions that are appropriately drawn based on the data presented.

Reviewer #1: Yes

Reviewer #2: Yes

3. Has the statistical analysis been performed appropriately and rigorously?

Reviewer #1: N/A

Reviewer #2: Yes

4. Have the authors made all data underlying the findings in their manuscript fully available (please refer to the Data Availability Statement at the start of the manuscript PDF file)?

Reviewer #1: (No Response)

Reviewer #2: Yes

5. Is the manuscript presented in an intelligible fashion and written in standard English?

Reviewer #1: Yes

Reviewer #2: Yes

6. Review Comments to the Author

Reviewer #1: Buis et al, R1

I thank the authors for their responses and revisions, most of my comments and concerns have been addressed.

A few issues that need attention (line numbers refer to track changes version):

98-99: Why "vaccines against prevention..."? I assume vaccines ae meant to prevent.

The sentence furthermore suggests that some of these vaccines are meant primarily as sterilizing vaccines (POI). I don't think that is the case. Some vaccines/candidates have been or are being evaluated against a POI endpoint, but that's something different.

105: Is this indeed the statement by Harris at al? The concerns about pre-vaccination testing to my knowledge are about vaccines that have only been licensed for use in IGRA-positive individuals, that's not POI.

199-200: my comment was not about duplicate articles about screening by two independent reviewers. I still believe the manuscript should state the percentage discrepancy in the 10% random sample so that the reader can be convinced that single review of the remaining 90% did not lead to major bias.

Reviewer #2: I am satisfied that all comments have been addressed adequately

7. PLOS authors have the option to publish the peer review history of their article (what does this mean?). If published, this will include your full peer review and any attached files.

**Do you want your identity to be public for this peer review?** For information about this choice, including consent withdrawal, please see our Privacy Policy.

Reviewer #1: **Yes: **Frank Cobelens

Reviewer #2: No

---

## [Decision Letter · Decision Letter 2]

11 Jul 2024

Mapping the Existing Body of Knowledge on New and Repurposed TB Vaccine Implementation: A Scoping Review

PGPH-D-24-00052R2

Dear Dr Joeri Sumina Sumina Buis,

We are pleased to inform you that your manuscript 'Mapping the Existing Body of Knowledge on New and Repurposed TB Vaccine Implementation: A Scoping Review' has been provisionally accepted for publication in PLOS Global Public Health.

Best regards,

Indira Govender

Academic Editor

Reviewer Comments (if any, and for reference):

Reviewer's Responses to Questions

**Comments to the Author**

1. If the authors have adequately addressed your comments raised in a previous round of review and you feel that this manuscript is now acceptable for publication, you may indicate that here to bypass the “Comments to the Author” section, enter your conflict of interest statement in the “Confidential to Editor” section, and submit your "Accept" recommendation.

Reviewer #1: All comments have been addressed

Reviewer #2: All comments have been addressed

2. Does this manuscript meet PLOS Global Public Health’s publication criteria? Is the manuscript technically sound, and do the data support the conclusions? The manuscript must describe methodologically and ethically rigorous research with conclusions that are appropriately drawn based on the data presented.

Reviewer #1: Yes

Reviewer #2: Yes

3. Has the statistical analysis been performed appropriately and rigorously?

Reviewer #1: N/A

Reviewer #2: I don't know

4. Have the authors made all data underlying the findings in their manuscript fully available (please refer to the Data Availability Statement at the start of the manuscript PDF file)?

Reviewer #1: Yes

Reviewer #2: Yes

5. Is the manuscript presented in an intelligible fashion and written in standard English?

Reviewer #1: Yes

Reviewer #2: Yes

6. Review Comments to the Author

Reviewer #1: All my remaining comments have been adequately addressed.

Reviewer #2: (No Response)

7. PLOS authors have the option to publish the peer review history of their article (what does this mean?). If published, this will include your full peer review and any attached files.

**Do you want your identity to be public for this peer review?** For information about this choice, including consent withdrawal, please see our Privacy Policy.

Reviewer #1: **Yes: **Frank Cobelens

Reviewer #2: No
